# Deep Feature Fusion and Optimization-Based Approach for Stomach Disease Classification

**DOI:** 10.3390/s22072801

**Published:** 2022-04-06

**Authors:** Farah Mohammad, Muna Al-Razgan

**Affiliations:** 1Department of Computer Science, College of Computer and Information Sciences, King Saud University, Riyadh 11451, Saudi Arabia; 2Department of Software Engineering, College of Computer and Information Sciences, King Saud University, Riyadh 11451, Saudi Arabia; malrazgan@ksu.edu.sa

**Keywords:** stomach disease, deep features, features optimization, disease classification

## Abstract

Cancer is the deadliest disease among all the diseases and the main cause of human mortality. Several types of cancer sicken the human body and affect organs. Among all the types of cancer, stomach cancer is the most dangerous disease that spreads rapidly and needs to be diagnosed at an early stage. The early diagnosis of stomach cancer is essential to reduce the mortality rate. The manual diagnosis process is time-consuming, requires many tests, and the availability of an expert doctor. Therefore, automated techniques are required to diagnose stomach infections from endoscopic images. Many computerized techniques have been introduced in the literature but due to a few challenges (i.e., high similarity among the healthy and infected regions, irrelevant features extraction, and so on), there is much room to improve the accuracy and reduce the computational time. In this paper, a deep-learning-based stomach disease classification method employing deep feature extraction, fusion, and optimization using WCE images is proposed. The proposed method comprises several phases: data augmentation performed to increase the dataset images, deep transfer learning adopted for deep features extraction, feature fusion performed on deep extracted features, fused feature matrix optimized with a modified dragonfly optimization method, and final classification of the stomach disease was performed. The features extraction phase employed two pre-trained deep CNN models (Inception v3 and DenseNet-201) performing activation on feature derivation layers. Later, the parallel concatenation was performed on deep-derived features and optimized using the meta-heuristic method named the dragonfly algorithm. The optimized feature matrix was classified by employing machine-learning algorithms and achieved an accuracy of 99.8% on the combined stomach disease dataset. A comparison has been conducted with state-of-the-art techniques and shows improved accuracy.

## 1. Introduction

In medical imaging, gastric lesion identification is an active research domain. Different gastric lesions include bleeding, esophagitis, ulcer, and polyps. Bleeding and ulcer are the most common among all abnormalities [1]. These stomach lesions have become a leading cause of mortalities in humans [2]. Globally, stomach cancer is the third major cause of death among all cancer deaths [3]. In the global cancer cases, esophageal cancer is at the seventh position [4]. In 2021 in the United States, Siegel et al. [5] estimated 26,560 new cases of stomach cancer of which 16,160 cases were in males and 10,400 were in females. The approximate deaths were 11,180, including 6740 males and 4440 females. In the United States, the American Cancer Society estimated there would be 26,380 new patients with stomach cancer for the year 2022. The estimated cases in men are 15,900 and 10,480 in women. The estimated deaths from these cases are 11,090, including 6690 males and 4400 females [6]. 

Wireless capsule endoscopy (WCE) [7] is the medical imaging method used to analyze the gastrointestinal (GI) tract. This imaging technique is widely utilized in clinics for the examination of the GI tract. In this technique, a tiny camera is sent into the GI tract to record the images. During the WCE procedure, about 50,000 images are generated. The gastroenterologists examine these images, which is a time-consuming process that is neither efficient nor very accurate. An average of two hours are required for an expert to analyze the images [8]. This issue can be resolved by developing automated computer-aided diagnostic (CAD) systems. These CAD models extract features from the WCE images to diagnose diseases. The fundamental steps of CAD systems include data acquisition, pre-processing, feature extraction, feature selection or feature optimization, and finally, classification.

In recent years, several computer vision (CV) researchers have introduced automated CAD models for the recognition of GI abnormalities [9,10,11,12]. The CAD models utilize handcrafted features and deep convolutional neural network (CNN) features for the recognition of WCE images. Different studies have utilized color features [13,14], texture features [15], point features [16], and histogram of oriented gradients (HOG) features [17]. With the recent developments in the deep-learning area, some researchers have utilized deep features to identify GI diseases. 

Stomach disease recognition is a challenging task due to the presence of several diseases and the availability of disease databases. The propound method of stomach disease classification’s main goal is to accurately predict the disease with less computation time. Deep-learning models inception v3 and DenseNet-201 are utilized in this paper for the deep feature extraction from stomach endoscopic images. Features fusion is performed on deep extracted features to attain rich feature space. The presence of redundant features which impacts the accuracy and prediction time came under consideration. The meta-heuristic approach was adopted to optimize the features for the accurate classification of stomach infections with less processing time. The robust selected features classification was performed using several machine algorithms. The major contribution of our propounded technique is as follows:Fine-tuning of pre-trained deep-learning models was performed with parameter modification for the derivation of high-level features by implementing the activation function at the average pooling layer.Deep features were concatenated using the parallel maximum covariance (PMC) method, and the resultant feature map was optimized using the nature-inspired modified binary dragonfly algorithm to reduce feature redundancy.

The remaining paper is organized as follows: Section 2 comprises related work, and Section 3 contains a brief description of the proposed methodology. Section 4 illustrates the experiments and results in a detailed description, and the conclusion of the paper is presented in Section 5.

## 2. Related Work

Several models have been designed for the detection and identification of stomach abnormalities. The presented techniques aim to identify stomach diseases in less computation time with higher accuracy. Different deep-learning techniques [18] equipped with feature optimization approaches for robust disease recognition have been utilized by the researcher [19]. In [20], the authors introduced a method for the recognition of bleeding and ulcers in WCE images. In the first step, they performed preprocessing techniques on the dataset including 3D-median filtering, 3D-box filtering, and HSV color variation. In the second phase, extraction of geometric features was performed to obtain binary images, and masks were generated and implemented to further improve the data. After that, the researchers extracted shape, and color, and speeded up robust features (SURF), and a serial-based fusion technique was applied to obtain the feature set. In the third step, feature selection was performed using the correlation coefficient and principal component analysis (PCA) approaches. Finally, in the classification phase, the researchers achieved a good classification accuracy on the SVM classifier. Zhao et al. [21] introduced TriZ, a rotation tolerant image feature for the recognition of gastric infections. Researchers compared the TriZ with the HOG method and observed a better recognition rate. Only 126 image features of TriZ were utilized for the experiment to compare with the HOG method. They achieved an accuracy of 87% using their model.

Majid et al. [22] classified stomach infections using handcrafted and deep features. They computed discrete cosine transform (DCT), color, discrete wavelet transforms (DWT), and VGG16 features in the feature extraction step. After extraction of the features, the feature set was given to the genetic algorithm (GA) for the selection of the best features. The maximum classification accuracy achieved on deep and classical optimized features was 96.5% on the ensemble classifier. Khan et al. [2] introduced an automated deep-learning technique based on deep feature classification for the recognition of stomach infections. First, researchers utilized a saliency-based technique for lesion detection. After lesion recognition, transfer learning was performed on the pre-trained VGG16 model, and features were computed from the fully connected layer. Then particle swarm optimization (PSO) was applied to the feature set for optimization. Finally, the selected features were fed to the classifiers for recognition. Using this model, they obtained the best accuracy on the Cubic SVM. 

Researchers in [23] recognized stomach diseases using deep features. Researchers extract features from the Inception V3 model and implemented PSO and the crow search algorithm (CSA) for feature optimization. The results of both optimization techniques were fused and fed into a multi-layer perceptron for classification. Their technique outperformed as compared to the other methods. Bora et al. [24] designed a model for the recognition of polyps. They utilized a generic Fourier descriptor (GFD) for shape features extraction, while color and texture features were collected using the non-subsampled contourlet transform (NSCT). After feature extraction, they evaluated the significance of features using the analysis of variance (ANOVA). In the feature selection phase, researchers utilized the fuzzy entropy-based method. Finally, the selected features were classified using a multi-layer perceptron, and least square SVM. Ayyaz et al. [25] utilized deep CNN models for the recognition of gastric abnormalities. First, they implemented the transfer-learning technique on AlexNet and VGG19. Then researchers extracted features from both CNN models and gave them to the GA for feature optimization. After that, the selected features were combined using the serial-based technique and given to the multiple classifiers for final recognition. Their model obtained a promising recognition rate on Cubic SVM. In [19], researchers extracted ResNet101 [26] features to classify stomach infections. Urban et al. [27] utilized features from VGG16 [28], VGG19 [29], and ResNet50 [26] models for the classification of polyps. Some researchers observed that the combination of the handcrafted and CNN features enhances the performance of the CAD model. Different classical and deep-learning-based approaches have been proposed by authors, but the challenges such as features redundancy, computational cost, and model robustness persist [30,31]. 

## 3. Materials and Methods

A detailed description of the proposed method is presented in this section. The method comprises several steps for stomach disease classification. In the first phase, data augmentation is employed to balance and increase the per-class images using flip and transpose operations. At the second phase, fine-tuned deep CNN models, namely inception v3 and DenseNet-201, are employed for deep features derivation by utilizing transfer learning and features fusion performed on deep derived features maps. The concatenated feature space has redundant features, which were removed using a metaheuristic approach. In the last phase, a features optimization binary dragonfly algorithm was implemented on the fused feature vector to remove redundant features, and classification was performed to classify the stomach disease. The detailed explanation of our proposed model for stomach disease classification is illustrated in Figure 1.

### 3.1. Data Acquisition and Preparation

The proposed stomach disease recognition method is utilized to classify the five major diseases of the stomach. We acquired two classes of bleeding and healthy from datasets [9] containing 3000 images per class. Esophagitis, polyps, and ulcerative colitis were extracted from publicly available challenging stomach gastrointestinal tract Kvasirv1 having 500 images per class and KvasirV2 having 1000 images per class [32]. The imbalance in the division of images per class and the varied sizes affected the performance of the proposed classification techniques. Data augmentation is a prominent step toward deep-learning performance. The inclusion of more data for the training of deep-learning models increases their performance. The maximum number of images per class is 3000, and the minimum images per class are 500. We performed different flip and transpose operations, which increased the number of images without losing the information and the features of images. The applied flip operation increased the dataset images and equaled out the images per class. The sample images after the augmentation process are shown in Figure 2. The mathematical notation of the flip operation on images is as follows: 

Let us have an input size of the image matrix 256×256 expressed as ${\tilde{N}}_{l,m}$ having lth rows and mth column and ${\tilde{N}}_{l,m}\in S^{l\times m}$. The matrix row $l=\left\{1,2,3\ldots \tilde{o}\right\}$ and column $m=\left\{1,2,3\ldots \tilde{p}\right\}$ where the number of channels is 3. The orientation of the RGB image was processed using three data augmentation operations.
(1)$${\tilde{N}}^{T}={\tilde{N}}_{l,m}$$

The transposition of the original image is expressed using ${\tilde{N}}^{T}$, the operation which alters the indices of images.
(2)$${\tilde{N}}^{H}={\tilde{N}}_{l}\left(\tilde{p}+1-m\right)$$

The horizontal flip of images is expressed as ${\tilde{N}}^{H}$.
(3)$${\tilde{N}}^{V}={\tilde{N}}_{\left(\tilde{o}+1-l\right)}$$

${\tilde{N}}^{V}$ illustrates the vertical image flip. The mentioned augmentation operation is repeated to equal the number of images per class and reach the 3500 number of images.

### 3.2. Deep Feature Extraction

Pattern recognition and computer vision tasks are performed using features derived from images. The features present an object according to its color, shape, and positioning. In the computer vision and medical imaging domain, the inclusion of deep learning significantly improves the diagnosis of diseases [33]. A deep convolutional neural network (CNN) normally has different layers such as the input layer, the convolutional layer, batch normalization, fully connected, and the activation layer ReLU. The CNN input layers feed the input data to the convolutional layer, and the weights are calculated. The activation function is applied using the ReLU layer, and inactive neurons are removed. The classification is performed on features computed using a fully connected layer by SoftMax layer. In our proposed method of stomach disease recognition, two deep CNN models, inception v3 and DenseNet-201 were implanted for deep feature derivation. The next sections comprise a description of the deep-learning models.

#### 3.2.1. Inception v3

Inception v3 is a deep-learning model that has robust performance in classification tasks. The model has a directed acyclic graph (DAG) with 94 convolutional layers, 316 layers, and 350 connections [34]. The architecture of the DAG network is complex due to multiple inputs to different layers at the same time. Several masks are implemented on different layers for the derivation of different features. The diverse architecture of inception v3 allows employing different masks and parameters on different layers as compared to the conventional CNN model, which has predetermined parameters on layers. Inception v3 was trained on ImageNet, a challenging image dataset having over a million images and 1000 classes [35]. The deep-learning model learned information about multiple objects and categories. The input size of inception v3 was 299×299×3. The first convolutional layer was processed by performing activation to derive a feature matrix having a size of 149×149×3, and the filter applied had a size of 32. The ReLU function was utilized to perform activation, and the next batch normalization was performed. ReLU, an activation function, is expressed as follows:(4)$$Te_{j}^{N}=\max\left(lw.lw^{\left(N-1\right)}\right)$$

A pooling layer is sandwiched in convolutional layers having a filter size of 2×2, which is illustrated below
(5)$$oz_{1}^{\left(q\right)}=oz_{1}^{\left(q-1\right)}$$
(6)$$oz_{2}^{\left(q\right)}=\frac{oz_{2}^{\left(q\right)}-H\left(q\right)}{S^{q}}+1$$
(7)$$oz_{3}^{\left(q\right)}=\frac{oz_{3}^{\left(q\right)}-H\left(q\right)}{S^{q}}+1$$
where $oz_{1}^{\left(q\right)}$, $oz_{2}^{\left(q\right)}$, and $oz_{3}^{\left(q\right)}$ are filters applied on the feature vector, and $S^{q}$ expresses the stride. Several layers are concatenated before the average pooling layer. The average pooling layer deployed for the derivation of the deep CNN feature map has a size $FV_{1}\times 2048$ by performing activation.

#### 3.2.2. DenseNet-201

A deep CNN model is known for the robust classification and recognition tasks. The DenseNet-201 [36] layers, having a directed connection to the next layers, increase the learning rate with minimal information loss. The network has fewer parameters in comparison to the other CNN models. Information maintainability requires less information loss from the first to the last layer. The information and features extracted at the different network layers can easily be predicted. The presence of gradient function decreases the chances of overfitting. The input size of the DenseNet-201 is 224×224×3. 

Suppose an image $\xi_{z}$ is used as an input to DenseNet-201. The network has M number of layers and transformation filters which are nonlinear $S_{m}\left(.\right).$ The transformation filter $S_{m}\left(.\right)$ is utilized as a concatenated function of convolution, batch normalization pooling, and ReLU. The output of the $m^{th}$ the layer is computed in a classical CNN model as ${\left(m+1\right)}^{th}$ and mathematical modeling as follows:(8)$$B_{m}=S_{m}\left(B_{m}-1\right)$$

DenseNet-201 layers have a direct connection to each other, and the $m^{th}$ layers carry information computed from all network layers $b_{0},b_{1}\ldots \ldots b_{n-1}$ which can be defined as
(9)$$b_{n}=S_{m}\left([b_{0},b_{1}\ldots \ldots b_{n-1}\right])$$

In the above equation, $b_{0},b_{1}\ldots \ldots b_{n-1}$ are the computed feature map of layer 0,……n−1. The resultant feature vector of previous layers is utilized by the average pooling layer and activation is employed to extract the required deep CNN feature map having a size of FV×1920.

#### 3.2.3. Feature Extraction Using Transfer Learning

Transfer learning has been employed on the deep CNN model for different machine-learning tasks and proved to be robust in recognition and classification tasks. We implanted transfer learning on our pre-trained fine-tuned deep CNN models for deep features extractions. The created stomach disease classification data was used to collect deep features by adopting transfer learning. We utilized 70:30 practice for the data split, training was performed on 70% data, and the remaining 30% was used for testing. The preprocessing step resized the images according to the input size of fine-tuned deep-learning networks. We employed the fine-tuned DenseNet-201, and the convolutional layer of the network was utilized as an input layer. The average pooling layer was used for the implementation of the activation function for features computation. The derived features map had a FV×1920 size and s expressed as $\varphi^{m1}$. The architecture of DenseNet for transfer learning is expressed in Figure 3.

We employed the pre-trained Inception V3 for deep features derivation by utilizing transfer learning. The convolutional layer used for image input and average pooling layer was activated using the activation function. The derived features vector size was $FV_{1}\times 2048$ and the feature map of derived features are illustrated using $\varphi^{m2}$. The derived features from DensNet-201 and inception V3 were fused for training and testing of the model. The architecture of inception v3 is illustrated in Figure 4. Training of the pre-trained model was performed using a sigmoid function. We utilized different parameters such as 250 epochs, iteration 30, learning rate 0.0001, batch size 64, and data shuffling was performed on all epochs.

### 3.3. Features Fusion

Feature fusion provides rich feature space for pattern recognition applications [37]. Recognition and classification of objects and images require an enriched feature space to recognize the specific image and feature concatenation that provides specific dense feature space. Feature concatenation provides efficient feature space for the classification of objects but impacts the processing time. A novel feature concatenation technique, parallel maximum covariance (PMC), is employed in our method for feature fusion. The integration of two deep CNN feature maps results in a single feature space.

Suppose the deep CNN features extracted from two deep CNN models are $\varphi^{m1}$ and $\varphi^{m2}$. The dimensions of these two feature vectors Fv and Fv1 are q×r and q×s, the q represents the number of images in a feature space where r and s represents the length of the attribute matrix. The features extracted using the pre-trained inception v3 are q×2048, and the features derived using the pre-trained deep CNN model DenseNet-201 are q×1920. The dimensions of feature spaces are equalized by computing the average and the addition of the average as padding. Suppose the d denotes a column vector n in a feature space $\varphi_{1}$, and pattern field $\varphi_{2}$ has a column vector e. The utilization of time series presents the row vectors as:(10)$$z_{1}=\varphi_{1}{}^{T}\varphi^{m1}$$
(11)$$z_{2}=\varphi_{2}{}^{T}\varphi^{m2}$$

The maximum covariance of $\varphi_{1}$ and $\varphi_{2}$ can be described as
(12)$$\overset{ˇ}{f}=Cov\left[z_{1},z_{2}\right]$$
(13)$$\overset{ˇ}{f}=Cov\left[\varphi_{1}{}^{T}\varphi^{m1},\varphi_{1}{}^{T}\varphi^{m1}\right]$$
(14)$$\overset{ˇ}{f}=\frac{1}{n-1}Cov\left[\varphi_{1}{}^{T}\varphi^{m1},\varphi_{1}{}^{T}\varphi^{m1}\right]$$
(15)$$\overset{ˇ}{f}=\varphi_{1}\left(F\varphi_{1}\varphi_{2}\right)\;\varphi_{2}$$
(16)$$F\varphi_{1}\varphi_{2}=\frac{1}{n-1}\left(\varphi^{\left(m1\right)}\varphi^{\left(m2\right)T}\right)$$
where $F\varphi_{1}\varphi_{2}$ represents covariance between $\varphi_{1}$ and $\varphi_{2},$ and features are expressed as ith and jth for $\varphi_{i}$ and $\varphi_{j}$. The maximum covariance of the final concatenated feature space is $F\varphi_{1}\varphi_{2}$. The features concatenation process makes the feature map dense and, in addition, creates redundant features. The final feature map obtained after feature fusion has a $FV_{2}\times 3968\;$ dimension.

### 3.4. Features Optimization

Feature assortment is the robust process of obtaining the most relevant features using features selection algorithms. The prime goal of features optimization is to reduce the presence of irrelevant features that impact classification functioning and computation time. The implanted features optimization method is illustrated in Figure 5. In our propounded technique, we utilized the binary dragonfly metaheuristic optimization method which utilized the KNN fitness function for robust feature selection. A detailed description of the implanted optimization method dragonfly is presented here.

Recently, Mirjalili et al. [38] introduced the nature-inspired dragonfly algorithm (DA). This is a population-based metaheuristic technique, motivated by the hunting and migration behavior of dragonflies. Small groups of dragonflies move to find food sources in what is known as the hunting procedure. In the migration process, the larger groups of dragonflies fly in one direction. The swarming behavior of dragonflies is explained by the following five parameters.

Separation: This operator ensures that the search agents in the neighborhood keep away from each other. Mathematically, it can be described as:(17)$$P_{i}=-\sum_{j=1}^{N}A-A_{j}$$
where N represents the neighborhood size, A is the current location of the individual, and $A_{j}$ denotes the *j*-th neighbor of the position A.

Alignment: This parameter describes the velocity of an individual according to the other neighboring individuals. The following equation describes this behavior:(18)$$X_{i}=\frac{\sum_{j=1}^{N}W_{j}}{N}$$

$W_{j}$ is the velocity of the *j*-th neighbor.

Cohesion: It indicates the individual’s movement behavior between the neighborhood to the center of the mass. This can be mathematically explained as:(19)$$M_{i}=\frac{\sum_{j=1}^{N}A_{j}}{N}-A$$

Attraction: This parameter defines the attraction of the flying individual towards a food source. This can be modeled as:(20)$$H_{i}=H_{loc}-A$$
where $H_{loc}$ denotes the location of the food source.

Distraction: The movement of an individual away from the enemy is a distraction. It can be given as:(21)$$K_{i}=K_{loc}+A$$
where, $K_{loc}$ is the enemy position.

In the DA, the optimization problem is solved using a step vector and a position vector. The following equation defines the step vector:(22)$$\Delta A_{t+1}=\left(sS_{i}+xX_{i}+mM_{i}+hH_{i}+kK_{i}\right)+\omega \Delta A_{t}$$
where s is the separation weight, $S_{i}$ denotes the *i*-th individual separation, x denotes the alignment weight, $X_{i}$ represents the alignment of *i*-th individual, m is the cohesion weight, $M_{i}$ represents the cohesion of i-th individual, h is the food factor, $H_{i}$ denotes the *i*-th individual’s food source, k is the enemy factor, $K_{i}$ is the location of the enemy of the *i*-th individual, ω indicates the inertia weight, and t indicates the iteration number which is 100 in our proposed modified optimization model.

In the search space, a step vector is added to the previous position to revise the position of dragonflies. However, the following equation is used in a binary search space:(23)$$A\left(t+1\right)=\left\{\begin{array}{c} A_{t},r<T\left(\Delta a_{t+1}\right)\;\\ A_{t},r\geq \;T\left(\Delta a_{t+1}\right) \end{array}\right.$$
where r represents the random number between the range 0 and 1. $T\left(\Delta a_{t+1}\right)$ is the transfer function that calculates the probability of a location update for all dragonflies, and this is given as:(24)$$T\left(\Delta a\right)=\left|\frac{\Delta a}{\sqrt{\Delta x^{2}+1}}\right|$$

The pseudo-code of the modified Binary Dragonfly is presented below (Algorithm 1):
**Algorithm 1:** Pseudocode of the Dragonfly Algorithm (DA)

Input: Feature Vector (
N×3968
)
Output: Selected Feature vector (
N×1855
)
Maximum Iteration = 100
Step 1: Initializing the population
$A_{i}\left(i=1,\;2,\;3,\;\ldots ,\;n\right)$
Step 2: Initialize
$\Delta A_{i}\left(i=1,\;2,\;3,\;\ldots ,\;n\right)$
Step 3: while (
t<Max Iteration
) do
-evaluate each dragonfly-Update $\left(H\right)\;and\;\left(K\right)$-Update the coefficients $\left(s,\;x,\;m,\;h,\;k,\;and\;\omega \right)$-Calculate $\left(S,\;X,\;M,\;H,\;and\;K\right)$-Update Step Vector $\left(\Delta A_{t+1}\right)$-Update $A_{t+1}$
Step 4: Return: the best solution


## 4. Results

The proposed technique of stomach disease classification is implanted on the stomach dataset created from challenging datasets. We acquired different disease images such as a polyp, ulcerative-colitis, and esophagitis from challenging datasets such as Kvasir V1 and Kvasir V2 [32] and two classes, bleeding, and healthy obtained in [9]. A few sample images from the created dataset are shown in Figure 6. Cross-validation of the 10-fold and the 70:30 ratio was employed, 70% for training and 30% for testing. 

The extensive experiments and simulation of the propounding method were processed on a system having an Intel Core i7 9th generation processor, a memory of 16 gigabytes, and 11 gigabytes of the graphic processing unit. Several machine-learning algorithms were employed for the prediction of stomach diseases, and a robust one was determined based upon accurate disease prediction and processing time. The robustness of the implanted method was assessed using several performance validation metrics such as accuracy, precision, recall, f1-score, false-negative rate (FNR), and computational time.

### 4.1. Deep Feature Fusion Results

A brief description of the aspired methodology results has been demonstrated in this section. The numerical outcomes of the implanted deep features fusion technique are shown in Table 1. The deep features are derived from pre-trained deep CNN models and parallel fusion functions. Several machine-learning classifiers were applied to the fused feature map to perform the stomach disease classification. In our experiments, the cubic SVM obtained the highest recognition accuracy of 96.2% with an FNR of 3.8%. The remaining machine-learning algorithms employed for the recognition of stomach disease on the fused feature space were linear support vector machine (L-SVM), quadratic support vector machine (Q-SVM), medium Gaussian support vector machine (MG-SVM), cosine Gaussian support vector machine (CG-SVM), Gaussian-naïve Bayes (GN-Bayes), ensemble subspace discriminant (ESD), cosine K-nearest neighbor (C-KNN), and linear discriminant (LD) and their attained corresponding classification results are 96%, 96.1%, 95.6%, 95.2%, 93.7%, 90.7%, 93.6%, 91.5%, respectively. The concatenation of deep CNN features increased the accurate classification but also increased the processing time.

### 4.2. Deep Feature Optimization Results on (CV = 10)

The main goal of the proposed technique was to accomplish the highest classification accuracy of stomach diseases in minimal computational time. The obtained fused feature space was optimized using a robust feature optimization algorithm called the binary dragonfly algorithm. The optimization technique eliminated the redundant features and picked the most relevant and robust features. The selection of the best features increases the disease classification functioning in less processing time. The proposed features optimization technique outcomes are presented in Table 2. Different machine algorithms are adopted for stomach disease recognition, different performance evaluation measures are utilized for the assessment of the methodology, and a robust one was selected based on processing time and accurate classification rate.

C-SVM attained the highest accuracy of 99.8% with 0.2% FNR, 99.8% precision, 99.4% recall, and 99.6% f1-score in a 33.18 s processing time. L-SVM, Q-SVM, MG-SVM, CG-SVM, GN-Bayes, ESD, C-KNN, and LD accomplished an accuracy of 98.6%, 99.1%, 96.5%, 95.6, 92.6%, 94.2%, 97.4%, and 98.3%, respectively. The most robust classifier in terms of processing time was LD with a processing time of 32.14 s, and the worst classifier was MG-SVM with a computational time of 252.343 s as presented in Figure 7.

### 4.3. Deep Feature Optimization Results on (CV = 15)

This section of results provides the proposed feature fusion and optimization method results on several machine-learning classifiers. Different optimized features vectors were fed to machine-learning algorithms to evaluate the performance of the proposed fusion and optimization approach. The main goal of experiments of the proposed technique at (CV = 15) was to analyze the performance of stomach disease recognition. Results are presented in Table 3 and show the highest accuracy of 99.6% with a 99.3 f1 score. 

In addition to the highest accuracy, the other performance evaluation measure such as precision 99.2%, recall 99.4% with 0.4% FNR were also computed on the C-SVM machine-learning algorithm. The worst classification performance was achieved on GN-Bayes with 98.3% accuracy. The best machine algorithm which utilized less computational cost was linear discriminant, and the worst classifier in terms of computation time was MG-SVM. The results show that the C-SVM achieved overall better accuracy, and LD achieved the best computational time. The overall performance of C-SVM is better in stomach disease recognition.

The results highlight the robustness of the proposed deep features extraction and optimization using the binary dragonfly algorithm. The highest classification accuracy of machine-learning algorithms using features optimization also reduced the computational time. The computational time of employed classifiers is presented in Figure 7. 

The optimized feature map classification was executed using various machine-learning algorithms, and a robust one was selected based upon the highest accuracy and less processing time. The C-SVM obtained the highest classification accuracy of 99.8% with a FNR of 0.2%. The robustness of the C-SVM classifier was also verified using the confusion matrix expressed in Table 4. 

The proposed deep CNN-based stomach disease classification methodology compared with other deep CNN models such as AlexNet, VGG19, VGG16, InceptionV3, and GoogleNet utilized for stomach disease classification and recognition accuracy has been presented in Figure 8. We have also trained our fine-tuned models and performed classification of stomach disease and compared with them. The results show that the pre-trained model accuracy of AlexNet, VGG16, VGG19, GoogleNet, and Resnet50 was 94.67%, 93.96%, 95.9%, 97.5, and 92.17, respectively, while the proposed model attained an accuracy of 99.8%. 

## 5. Discussion

In this work, we performed a fair comparison of our proposed methodology with other pertinent techniques presented for stomach disease recognition. In our proposed features extraction and optimization method, we utilized cross-validation of 10-fold to examine the robustness of our methodology. The maximum accuracy achieved by concatenating deep features spaces was 96.2%, and the deployment of a modified dragonfly algorithm increased the accuracy to 99.8 using C-SVM machine-learning classifiers. The comparison of the proposed feature optimization method with relevant methodologies is presented in Table 5. The techniques that are used for the comparison of the proposed method used the same dataset or same classes of stomach dataset used in this proposed work. Researchers [39] utilized 12,147 stomach disease endoscopic images to extract from Kvasir v2 and endoscopy artifact detection (EAD) [40] for disease recognition, using a deep-learning-based attenuation technique and achieved a classification accuracy of 93.19%. In [41] stomach disease recognition was performed on 6702 images from Kvasir V2, challenging the stomach disease dataset using data augmentation and fine-tuning of CNN models for stomach disease classification with an accuracy of 96.33%. In [42] 2006, capsule endoscopy images acquired from Kiang Wu Hospital used by researchers for the classification of gastric disease using a deep attention model for segmentation to locate the lesion region for accurate recognition of disease recognition and attained 96.76% of stomach disease classification accuracy. In [43] researchers employed CNN and capsule network for stomach disease classification and deformation analysis using a Kvasir V2 dataset having 8000 images with an accuracy of 94.73%. In [44] the authors classified ulcerative colitis from challenging datasets Kvasir, Kvasir V2, and hyper-Kvasir for binary classification using deep learning and achieved an accuracy of 87.50%. Our proposed deep feature extraction and optimization technique for stomach cancer classification was applied on 17,500 images of different stomach disease classes for stomach diseases recognition from Kvasir and Kvasir v2 and two classes of healthy and bleeding obtained from [9] and achieved the highest accuracy of 99.8% on the. cubic SVM classifier. The proposed model also reduced the computational time as compared to modern techniques of stomach infection recognition. We performed experiments on the fused feature vector and compared these results after applying the optimization method. The results clearly show that the proposed binary dragonfly algorithm increases the efficiency of the model as presented in Figure 9.

The robustness and consistency of the proposed method were also evaluated using a confidence interval, a statistical method used for the computation of error and continuity exemplified in Figure 10. Normally, in data representation, 5% is used as a statistical significance for the confidence level score of 95% with a standard deviation of 0.08. The presence of error in our proposed method is 99.74 ± 0.118 (±0.12%). The statistical analysis presented in the figure shows that the proposed stomach disease classification method is consistent and accurate at different confidence intervals.

## 6. Conclusions

Deep CNN-based feature derivation and optimization methodology have been presented in this paper for stomach disease classification. In the suggested method, the deep CNN features derivation was performed by employing transfer learning using Inception V3 and DenseNet-201, Activation was performed on the feature derivation layers of both models to obtain the specific feature matrix. The derived feature matrix was fused with the help of the parallel maximum covariance method. The maximum accuracy achieved after feature fusion was 96.2% on the C-SVM. The concatenated features matrix was optimized using a modified binary dragonfly algorithm. The features optimizing technique provided better results as compared to the individual deep CNN feature fusion feature map. The maximum recognition accuracy of stomach disease recognition after feature optimization was 99.8% on a C-SVM machine-learning classifier which represents the robustness of the proposed feature optimization technique. The main strength of the proposed method comprised robust feature derivation and selection to enhance the accuracy of stomach diseases recognition. The major disadvantage of the implanted technique is the increase in computational cost due to feature concatenation. Future work will comprise the large database creation and optimal deep-learning model creation, especially for stomach disease recognition. Furthermore, a deep-learning model will be trained from scratch to perform polyp and ulcer segmentation.

## Figures and Tables

**Figure 1 sensors-22-02801-f001:**
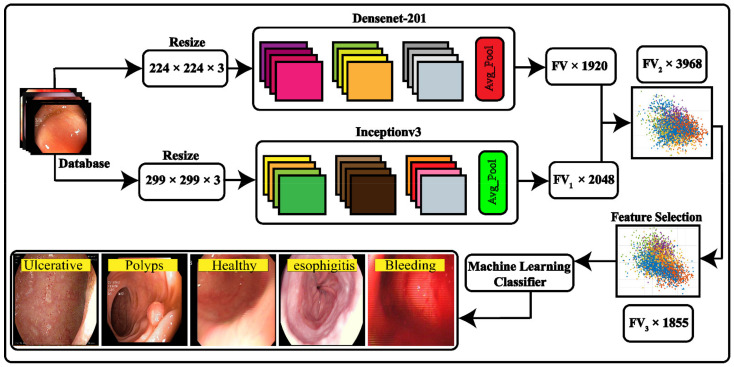
Proposed technique for stomach disease classification.

**Figure 2 sensors-22-02801-f002:**
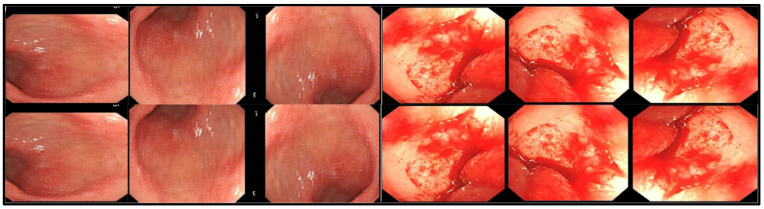
Sample Images after data augmentation.

**Figure 3 sensors-22-02801-f003:**
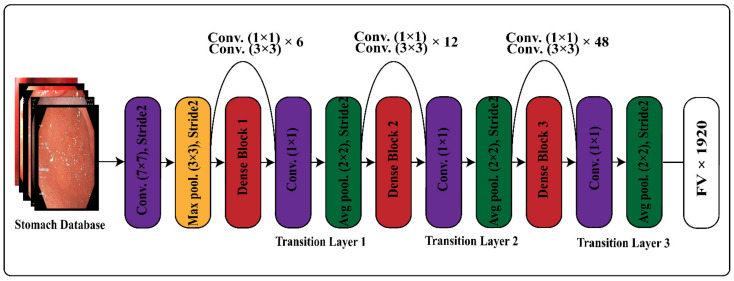
Transfer-learning architecture of DenseNet-201 for feature extraction.

**Figure 4 sensors-22-02801-f004:**
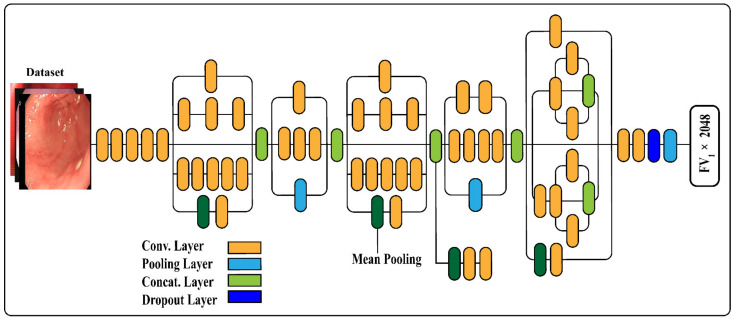
Transfer-learning formation of Inception V3 for feature derivation.

**Figure 5 sensors-22-02801-f005:**
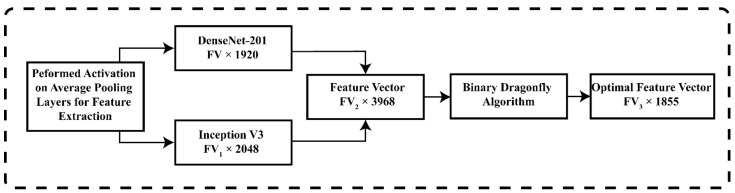
Proposed feature optimization architecture.

**Figure 6 sensors-22-02801-f006:**
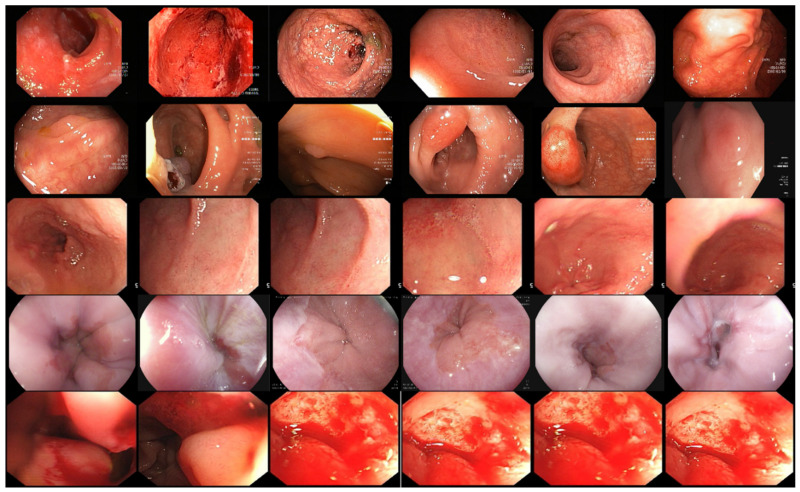
Sample images from stomach disease classification dataset.

**Figure 7 sensors-22-02801-f007:**
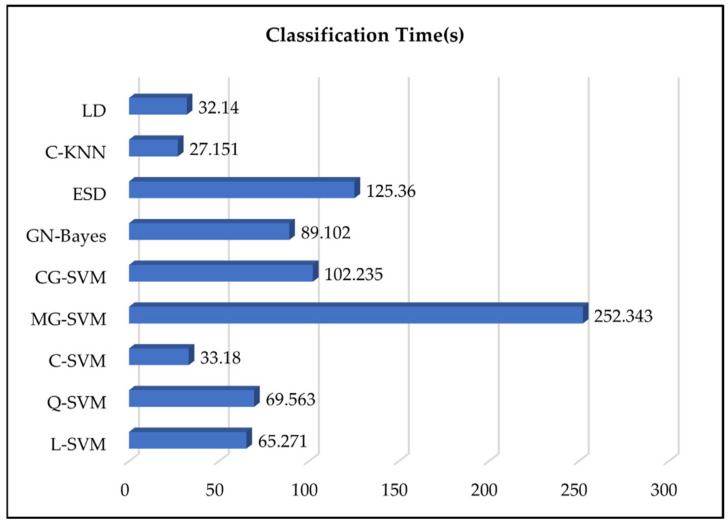
Computational time comparison of utilized machine-learning classifiers.

**Figure 8 sensors-22-02801-f008:**
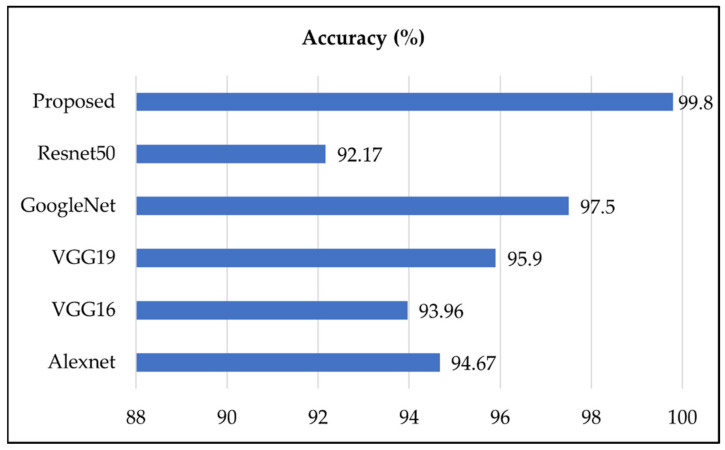
Proposed stomach disease recognition model comparison with CNN models.

**Figure 9 sensors-22-02801-f009:**
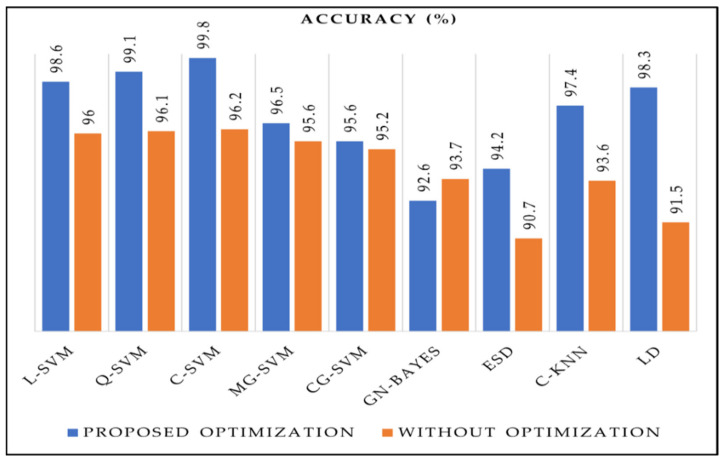
Stomach disease recognition accuracy with and without optimization.

**Figure 10 sensors-22-02801-f010:**
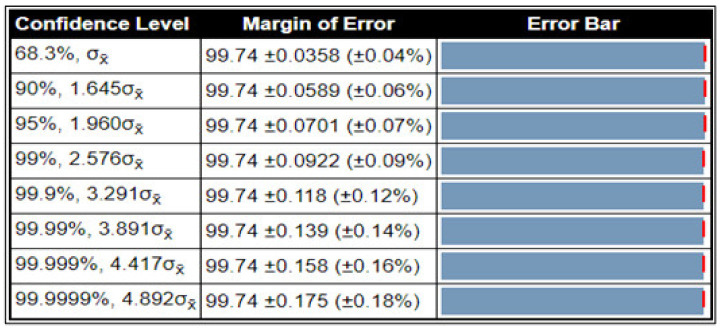
Statistical confidence interval of proposed stomach classification method.

**Table 1 sensors-22-02801-t001:** Stomach disease classification results using deep CNN features fusion.

Classifier	Evaluation Measures		
	Accuracy (%)	FNR (%)	Time (s)
L-SVM	96	4	514
Q-SVM	96.1	3.9	730
C-SVM	96.2	3.8	611
MG-SVM	95.6	4.4	1310
CG-SVM	95.2	4.8	1214
GN-Bayes	93.7	6.3	354
ESD	90.7	9.3	125
C-KNN	93.6	6.4	153
LD	91.5	8.5	272

**Table 2 sensors-22-02801-t002:** Stomach disease classification with proposed feature optimization technique (CV = 10).

Classifier	Evaluation Measures					
	Accuracy (%)	Precision (%)	FNR (%)	Recall (%)	F1_Score (%)	Time(s)
L-SVM	98.6	98.8	1.4	99.1	98.9	65.271
Q-SVM	99.1	99.2	0.8	98.3	98.7	69.563
C-SVM	**99.8**	**99.8**	**0.2**	**99.4**	**99.6**	**33.18**
MG-SVM	96.5	97	3.5	97.5	97.2	252.343
CG-SVM	95.6	95.6	4.4	96.6	96.1	102.235
GN-Bayes	92.6	92.7	7.3	93.7	93.2	89.102
ESD	94.2	94.3	5.8	95.2	94.7	125.36
C-KNN	97.4	98.4	2.6	98.4	98.4	27.151
LD	98.3	98.7	1.7	99.3	99	32.14

**Table 3 sensors-22-02801-t003:** Stomach disease classification with proposed feature optimization technique (CV = 15).

Classifier	Evaluation Measures					
	Accuracy (%)	Precision (%)	FNR (%)	Recall (%)	F1_Score (%)	Time(s)
L-SVM	99.5	98.4	0.5	99.1	98.7	67.271
Q-SVM	99.5	99.1	0.5	98.3	98.7	73.561
C-SVM	**99.6**	99.2	0.4	99.4	99.3	39.18
MG-SVM	99.4	99.3	0.6	99.5	99.4	286.393
CG-SVM	99.2	99	0.8	99.4	99.2	152.135
GN-Bayes	98.3	98.1	1.7	99.3	98.7	92.112
ESD	99.2	99.1	0.8	99.4	99.2	139.36
C-KNN	99.4	98.6	0.6	99.6	99.1	**29.121**
LD	98.9	98.4	1.1	99.3	98.8	35.11

**Table 4 sensors-22-02801-t004:** Confusion matrix of proposed features optimization for stomach disease classification.

Stomach Disease	Stomach Diseases				
	Healthy	Bleeding	Esophagitis	Polyps	Ulcerative-Colitis
Healthy	100%				
Bleeding		100%			
Esophagitis			100%		
Polyps			<1%	99%	<1%
Ulcerative-Colitis				<1%	99%

**Table 5 sensors-22-02801-t005:** Comparison of the proposed model with existing techniques.

Ref.	Year	Dataset	No. of Images	Accuracy
[39]	2021	Kvasir V2 + EAD2019	12,147	93.19%
[41]	2021	Kvasir V2	6702	96.33%
[42]	2022	Kiang Wu Hospital dataset	2006	96.76%
[43]	2022	Kvasir V2	8000	94.83%
[44]	2022	Kvasir + Kvasir V2	3482	87.50%
**Proposed**	**2022**	Kvasir + Kvasir V2 + Healthy + Bleeding	**17,500**	**99.8%**

## Data Availability

The dataset is publicly available and can be used for research and education purpose. The dataset is available at: The Kvasir Dataset (simula.no).

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
