# Peer review of "Deep Feature Fusion and Optimization-Based Approach for Stomach Disease Classification"

_sensors, 2022, doi:10.3390/s22072801_

Round 1

Reviewer 1 Report

The paper proposes a deep learning-based stomach disease classification method using deep features extraction, fusion, and optimization. The results of experiments on a benchmark dataset of Wireless Capsule Endoscopy (WCE) images are presented. The paper needs to be improved by addressing methodological and presentation issues before it could be considered for publication.

Comments:

  1. My biggest concern is about novelty. The study uses some of the known deep learning models (two pre-trained deep CNN models Inception v3 and DenseNet-201), which have been used widely before for various biomedical image analysis and classification tasks with various optimizations and modifications. What is exactly your specific innovation? The list of the contributions at the end of the Introduction section lists some common tasks of deep learning based workflow such as data augmentation and finetuning, but find nothing innovative. Clearly state your novelty and difference from the previous works.
  2. The related works section is rather disappointing. I would expect an introductory paragraph explaining the classification (or grouping) of previous works to be discussed. Now the related papers are introduced without any particular order and the selection of works to discuss seems to be ad hoc. The style of presentation is poor (someone did something) and lacks of essential details needed to understand the contribution and findings of the analyzed works. Some of the recent works seem to be missed from the discussion such as “StomachNet: Optimal deep learning features fusion for stomach abnormalities classification”, and “Stomach Diseases Classification: A Framework of Deep Neural Network Robustness against Adversarial Attack in WCE Images”. Finally, there is no summary of the results of the literature analysis. What is the outcome of this section? You need to identify the limitations of previous works, which may serve as a motivation of the current study.
  3. How do you deal with the class imbalance in the dataset? Did you do some balancing?
  4. Clearly specify the parameters of the image augmentation operations used for replicability of this study.
  5. From which layers of neural network models did you take the deep features?
  6. Why did you select the Dragonfly algorithm for feature optimization? Where are many other nature-inspired heuristic algorithms available? Did you consider other algorithms? Provide your motivation.
  7. Perform the statistical analysis of the results. Calculate standard deviation or 95% confidence limits of the performance measures using the results from all classification folds.
  8. Present the hyperparameters of the final stage classifiers such as the number of nearest neighbours for KNN and, C and gamma for the linear SVM.
  9. Figure 10: scale error bar (for example to [90-100%]) for better visibility.
  10. Discuss the limitations of the proposed methodology.
  11. Improve the conclusions. Use the main numerical results from experiments to support your claims.

Author Response

Response to Reviewer 1 Comments

The paper proposes a deep learning-based stomach disease classification method using deep features extraction, fusion, and optimization. The results of experiments on a benchmark dataset of Wireless Capsule Endoscopy (WCE) images are presented. The paper needs to be improved by addressing methodological and presentation issues before it could be considered for publication.

Point 1: My biggest concern is about novelty. The study uses some of the known deep learning models (two pre-trained deep CNN models Inception v3 and DenseNet-201), which have been used widely before for various biomedical image analysis and classification tasks with various optimizations and modifications. What is exactly your specific innovation? The list of the contributions at the end of the Introduction section lists some common tasks of deep learning based workflow such as data augmentation and finetuning, but find nothing innovative. Clearly state your novelty and difference from the previous works.

Response 1: Authors are thankful to the honorable reviewers for such good comments. The major contributions of this work are enlisted at the end of the introduction section. Moreover, the main novelty of this work is the optimization of deep learning features.

Point 2: The related works section is rather disappointing. I would expect an introductory paragraph explaining the classification (or grouping) of previous works to be discussed. Now the related papers are introduced without any particular order and the selection of works to discuss seems to be ad hoc. The style of presentation is poor (someone did something) and lacks of essential details needed to understand the contribution and findings of the analyzed works. Some of the recent works seem to be missed from the discussion such as “StomachNet: Optimal deep learning features fusion for stomach abnormalities classification”, and “Stomach Diseases Classification: A Framework of Deep Neural Network Robustness against Adversarial Attack in WCE Images”. Finally, there is no summary of the results of the literature analysis. What is the outcome of this section? You need to identify the limitations of previous works, which may serve as a motivation of the current study.

Response 2: The related work has been updated in the revised manuscript. We added some latest articles in ascending order and summarized them at the end which shows the gap among them and the motivation of this work.

Point 3: How do you deal with the class imbalance in the dataset? Did you do some balancing?

Response 3: In this work, we performed a data augmentation step for handling imbalance issues.

Point 4: Clearly specify the parameters of the image augmentation operations used for replicability of this study.

Response 4: The steps performed have been highlighted in the manuscript. Horizontal and vertical flip operations were performed and image transpose was also taken.

Point 5: From which layers of neural network models did you take the deep features?

Response 5: The global average pooling layer opted for features extraction.

Point 6: Why did you select the Dragonfly algorithm for feature optimization? Where are many other nature-inspired heuristic algorithms available? Did you consider other algorithms? Provide your motivation.

Response 6: As compared to GA and PSO, the Dragonfly algorithm is performed well for medical data. Moreover, we computed results on each optimization algorithm and illustrated in Figure.

Point 7: Perform the statistical analysis of the results. Calculate standard deviation or 95% confidence limits of the performance measures using the results from all classification folds.

Response 7: the statistical analysis has been performed and changes have been incorporated in the manuscript.

Point 8: Present the hyperparameters of the final stage classifiers such as the number of nearest neighbours for KNN and, C and gamma for the linear SVM.

Response 8: In the future, we will also present discuss the hyperparameters of machine learning methods in the manuscript.

Point 9: Figure 10: scale error bar (for example to [90-100%]) for better visibility.

Response 9: figure 10 has been revised.

Point 10: Discuss the limitations of the proposed methodology.

Response 10: The limitations of this work have been added under the conclusion section.

Point 11: Improve the conclusions. Use the main numerical results from experiments to support your claims.

Response 11: As per recommendation, the conclusion section has been updated in the revised manuscript.

Reviewer 2 Report

Dear authors, the submitted manuscript is well conceived, in-depth, informative and scientifically structured.   However, I would like to add a few suggestions that might be implemented to modify the manuscript for better quality control.  

1. Please include all the Software codes that have been implemented in this particular research work in the appendix section for independent simulation, testing, validation and integration.  
2. Only a few references have been included in the manuscript.  Please add relevant references to enhance the global importance of the paper.  
3. If possible, it will be good if the authors could add a graphical representation summarizing their results which compares controls, simulation results, all the model parameters and variables. 
4. Please, briefly add future perspectives and further applied applications of this specific research work in the discussion section before the conclusion.  
5. Please include and detail all the algorithms (mathematical expressions), of the related techniques and/or model/s mentioned in the manuscript.  
6. The techniques and/or models presented and mentioned in the manuscript require sufficient details (including calibration, sensitivity analysis and validation) to allow other researchers to develop and test the applications later on.  More simulations and comparisons that show the advantages and the drawbacks of the proposed schema are needed.

Author Response

Response to Reviewer 2 Comments

Dear authors, the submitted manuscript is well conceived, in-depth, informative, and scientifically structured.   However, I would like to add a few suggestions that might be implemented to modify the manuscript for better quality control.  

Point 1: Please include all the Software codes that have been implemented in this research work in the appendix section for independent simulation, testing, validation and integration.  

Response 1: Authors are thankful to the honorable reviewer for this comment. The entire code and datasets mat file will be available on ResearchGate after the acceptance of this paper.

Point 2: Only a few references have been included in the manuscript.  Please add relevant references to enhance the global importance of the paper.  

Response 2: As per recommendation, the latest articles related to the topic have been added to the revised manuscript.

Point 3: If possible, it will be good if the authors could add a graphical representation summarizing their results which compares controls, simulation results, all the model parameters and variables.

Response 3: We added confusion matrixes, time plots, and confidence interval-based graphical results in the revised manuscript.

Point 4: Please, briefly add future perspectives and further applied applications of this specific research work in the discussion section before the conclusion.  

Response 4: As per recommendation, future work has been added in the revised manuscript under the conclusion section.

Point 5: Please include and detail all the algorithms (mathematical expressions), of the related techniques and/or model/s mentioned in the manuscript.  

Response 5: The mathematical description of each step of the proposed methodology has been added in the revised manuscript.

Point 6: The techniques and/or models presented and mentioned in the manuscript require sufficient details (including calibration, sensitivity analysis and validation) to allow other researchers to develop and test the applications later on.  More simulations and comparisons that show the advantages and the drawbacks of the proposed schema are needed.

Response 6: As per the recommendation by the honorable reviewer, we added confidence interval-based results for further validation of the proposed scheme.

Reviewer 3 Report

The manuscript entitled "Deep Feature Fusion and Optimization Based Approach for Stomach Disease Classification" is interesting and will present interest for readers.

Abstract. I suggest to mention the originality of the study and the novelty it brings in the field, in order to point the difference other studies.

Several statements are repeated throughout the introduction section, this has to be avoided to
make the paper more comprehensive. It is well-known that common diseases and conditions such as diabetes, cancer or autoimmune diseases can be accelerated or delayed depending on dietary options. Thus, healthy choices must include foods with nutrients and bioactive compounds, which, beyond the nutritional properties, can be used as an effective prevention strategy.  It is important to state clearly implications for research, practice and society. In my opinion, for the audience, for those who are not working in this field, but have interested in this subject (nutritionists, education specialists, public health organizations, different organizations, policy makers,etc.), it is relevant to emphasize the importance of molecules from food in maintaining a balance health. In this regard, I kindly recommend the next paper to be consulting for the introduction section: DOI: http://dx.doi.org/10.5772/intechopen.91218.

The Experimental and Modeling Approach correctly. Results and Discussions could be improved by studying other papers in the field (just few similar studies was nominated).

Author Response

Response to Reviewer 3 Comments

The manuscript entitled "Deep Feature Fusion and Optimization Based Approach for Stomach Disease Classification" is interesting and will present interest for readers.

Abstract. I suggest to mention the originality of the study and the novelty it brings in the field, in order to point the difference other studies.

Response 1: Abstract of the manuscript has been revised as per recommendation.

Several statements are repeated throughout the introduction section, this has to be avoided to
make the paper more comprehensive. It is well-known that common diseases and conditions such as diabetes, cancer or autoimmune diseases can be accelerated or delayed depending on dietary options. Thus, healthy choices must include foods with nutrients and bioactive compounds, which, beyond the nutritional properties, can be used as an effective prevention strategy.  It is important to state clearly implications for research, practice and society. In my opinion, for the audience, for those who are not working in this field, but have interested in this subject (nutritionists, education specialists, public health organizations, different organizations, policy makers,etc.), it is relevant to emphasize the importance of molecules from food in maintaining a balance health. In this regard, I kindly recommend the next paper to be consulting for the introduction section: DOI: http://dx.doi.org/10.5772/intechopen.91218.

Response 2: As per recommendation, the introduction of the manuscript has been revised and added few latest works for the significance of this manuscript.

The Experimental and Modeling Approach correctly. Results and Discussions could be improved by studying other papers in the field (just few similar studies was nominated).

Response 3: The results section is further improved as per the recommendation. We added the ablation study and confidence interval-based analysis for the dominance of the proposed framework.

Reviewer 4 Report

This paper presents CNN-based feature derivation and optimization method for stomach disease classification. In the suggested method the deep CNN features derivation is performed by employing transfer learning using Inception V3 and DenseNet-201 and activation performed on the feature derivation layers of both models to obtain the specific feature matrix.. The topic is interesting and matches well for MDPI Sensors journal. The paper contains some review of related works. However the paper has some unclear points and the following minor and major concerns.

The authors describe the major contribution of the paper as follows:

«1. Data augmentation techniques flip, and transpose are utilized to increase the number of images per class without losing the information and class imbalance is removed.

2. Fine tuning of pretrained deep learning models performed with parameter modification for the derivation of high-level features by implementing activation function at the average pooling layer.

3. Deep features concatenated using parallel maximum covariance (PMC) method and resultant feature map optimized using nature inspired modified Binary Dragonfly algorithm to reduce the feature redundancy.»

1. The 1st contribution point is a rudimentary and generic approach to dataset augmentation. It is probably better to remove this point.

2. It is necessary to perform the ablation study for the proposed method.

What increase in classification accuracy does the use of “Fine tuning of pretrained deep learning models” give? How does the use of the PMC method affect the accuracy? How much does the modified Binary Dragonfly algorithm improve accuracy?

3. For works [37-40], which are given in Table 4, it is necessary to give at least a brief description of the methods that were used in them. For a correct comparison of the proposed method with the methods in [37-40], it is necessary to give not only the number of images in the dataset, but also describe these datasets. And also to get the result of the proposed method on those datasets that were used in works [37-40].

4. In my opinion, it makes sense to decipher the names of classifiers in

L372 - «L-SVM, Q-SVM, MG-SVM, CG-SVM, G-Naïve Bayes, ESD, ...»

Author Response

Response to Reviewer 4 Comments

This paper presents CNN-based feature derivation and optimization method for stomach disease classification. In the suggested method the deep CNN features derivation is performed by employing transfer learning using Inception V3 and DenseNet-201 and activation performed on the feature derivation layers of both models to obtain the specific feature matrix.. The topic is interesting and matches well for MDPI Sensors journal. The paper contains some review of related works. However the paper has some unclear points and the following minor and major concerns.

The authors describe the major contribution of the paper as follows:

«1. Data augmentation techniques flip, and transpose are utilized to increase the number of images per class without losing the information and class imbalance is removed.

  1. Fine tuning of pretrained deep learning models performed with parameter modification for the derivation of high-level features by implementing activation function at the average pooling layer.
  2. Deep features concatenated using parallel maximum covariance (PMC) method and resultant feature map optimized using nature inspired modified Binary Dragonfly algorithm to reduce the feature redundancy.»

  1. The 1st contribution point is a rudimentary and generic approach to dataset augmentation. It is probably better to remove this point.

Response 1: As per recommendation, the 1st point is removed from the revised manuscript under the part of the contribution.

  1. It is necessary to perform the ablation study for the proposed method.

What increase in classification accuracy does the use of “Fine tuning of pretrained deep learning models” give? How does the use of the PMC method affect the accuracy? How much does the modified Binary Dragonfly algorithm improve accuracy?

Response 2: A detailed ablation study is provided in the revised manuscript. Results are computed of each middle step that shows the importance of the proposed framework.

  1. For works [37-40], which are given in Table 4, it is necessary to give at least a brief description of the methods that were used in them. For a correct comparison of the proposed method with the methods in [37-40], it is necessary to give not only the number of images in the dataset, but also describe these datasets. And also to get the result of the proposed method on those datasets that were used in works [37-40].

Response 3: For each work given in the comparison table, used the same dataset that we utilized in this work.

  1. In my opinion, it makes sense to decipher the names of classifiers in

L372 - «L-SVM, Q-SVM, MG-SVM, CG-SVM, G-Naïve Bayes, ESD, ...»

Response 4: The full names of these classifies have been added in the revised manuscript like linear SVM (L-SVM), quadratic SVM (Q- SVM), etc. 

Round 2

Reviewer 1 Report

Improve the description of Algorithm 1: add line numbers, specify inputs and outputs, define the variables with their initial values and comment on their meaning, set the initial values of iterator t, specify the equations used to update and calculate values.

Author Response

Response to Reviewer 1 Comments

Round 2

Point 1: Improve the description of Algorithm 1: add line numbers, specify inputs and outputs, define the variables with their initial values and comment on their meaning, set the initial values of iterator t, specify the equations used to update and calculate values.

Response 1: As per the recommendations of the honorable reviewer, the changes have been incorporated into the manuscript. The equations have been incorporated and highlighted in the manuscript.

Input: Feature Vector ( )

Output: Selected Feature vector ( )

Maximum Iteration = 100

Step 1: Initializing the population

Step 2: Initialize

Step 3: while ( ) do

-          evaluate each dragonfly

-       Update

-       Update the coefficients

-       Calculate

-       Update Step Vector

-          Update

Step 4: Return: the best solution

Reviewer 2 Report

Dear Authors, I am okay with the changes made. Thanks. 

Author Response

Thanks to the honorable reviewer for your valuable comments.

Reviewer 4 Report

Unfortunately, I cannot recommend this paper for publication in the journal Sensors.

My following remark was ignored:

1. For works [37-40], which are given in Table 4, it is necessary to give at least a brief description of the methods that were used in them. For a correct comparison of the proposed method with the methods in [37-40], it is necessary to give not only the number of images in the dataset, but also describe these datasets. And also to get the result of the proposed method on those datasets that were used in works [37-40].

Authors response: For each work given in the comparison table, used the same dataset that we utilized in this work.

However, no description of the methods used for comparison appeared in the text. It is also not clear if the authors "for each work given in the comparison table, used the same dataset that we utilized in this work", why Table 5 (former Table 4) shows a different number of images in the data set for each method.

Author Response

Response to Reviewer 4 Comments

Round 2

  1. For works [37-40], which are given in Table 4, it is necessary to give at least a brief description of the methods that were used in them. For a correct comparison of the proposed method with the methods in [37-40], it is necessary to give not only the number of images in the dataset but also describe these datasets. And also to get the result of the proposed method on those datasets that were used in works [37-40].

Response 1: As per the recommendation of the honorable reviewer, the description of datasets used in relevant methods has been incorporated in the manuscript. The same dataset has been used in the manuscript. The methods used for comparison have used the same dataset but with a different numbers of images and classes.

However, no description of the methods used for comparison appeared in the text. It is also not clear if the authors "for each work given in the comparison table, used the same dataset that we utilized in this work", why Table 5 (former Table 4) shows a different number of images in the data set for each method.

Response 2: As per the recommendation of the honorable reviewer, a new article related to our work has also been added for comparison in the table. The same dataset has been used by the authors with which we have compared our work. The detailed description of the dataset and methods have been discussed in the manuscript and changes have been incorporated. The number of images has been mentioned in the table are same as mentioned by researchers in their manuscripts.

Round 3

Reviewer 4 Report

Accept in present form.

This manuscript is a resubmission of an earlier submission. The following is a list of the peer review reports and author responses from that submission.

Round 1

Reviewer 1 Report

The development of advanced AI models may lead to considerable technological improvements in the last years. In medicine, these models could be very helpful in disease detection.

The propose manuscript exposes a ‘Deep Feature Fusion and Optimization Based Approach for Stomach Disease Classification’ that addresses this problem. The manuscript is divided into five parts: an introduction, a part titled ‘Related works’ presenting similar research studies published in the literature, a third part on Materials and methods, a fourth part about the results and a conclusion.

Overall, the manuscript is well written and may be of interest to the readers of the journal. However, several clarifications are needed:

  • L 35 è 37: precise that these figures are projections
  • L 57-58: the research gaps and the novelty of your manuscript must be highlighted
  • L 106-108: precise the type of model
  • Fig 1is not clear at first sight, especially scatterplots before feature fusion as their names ‘N=1920’ and ‘M=2048’ are not given in context. Please revise these names and increase the figure resolution
  • L 149 – 150: flip and transpose-based augmentation is rather limited. Why did the authors not use rotation, contrast change?
  • L 167: What is the final amount of images all the various sets?
  • L 342-346: this part is rather repetitive and belong to Materials and methods section in my opinion
  • L 431: calculation time seems important. Can you give some figures on this calculation time?
  • Fig 9: error bar does not provide very interesting information in my opinion

Reviewer 2 Report

The quality of the figures is very poor.

In some figures, the resolution is quite low; in some figures the text is not even readable.

Reviewer 3 Report

In the abstract, the aim of the study should be better defined. Some information on the used methods should be also added.

In the Introduction, the part on the state of art should be reduced and more information on the innovation of authors' technology should be added.

Figure 1 in Materials&Methods should be reported in a better graphical resolution because it is not clear. The same for Figure 6.

In the Results section, a better discussion should be performed, in particular in relation to the previous technologies and methods validated for the stomach disease classification.

Conclusions should only report the main results of the study without numerical data. Please delete redundant information.

Round 2

Reviewer 1 Report

The comments are properly answered in my opinion.

Author Response

Thank you honorable reviewer for your valuable comments and for accepting my response.

Reviewer 2 Report

The Authors did not address the Reviewers'concerns in a satidfactory manner.

Moreover, the quality of the figures remains very low.

Author Response

Thanks, Honorable reviewer for your keen observations of the manuscript. The suggested changes have been incorporated into the manuscript.

High-quality figures have been incorporated into the manuscript by replacing low-quality figures.